# Study Protocol for an Online Questionnaire Survey on Symptoms/Signs, Protective Measures, Level of Awareness and Perception Regarding COVID-19 Outbreak among Dentists. A Global Survey

**DOI:** 10.3390/ijerph17155598

**Published:** 2020-08-03

**Authors:** Guglielmo Campus, Marcela Diaz-Betancourt, Maria Grazia Cagetti, Joana C. Carvalho, Thiago S. Carvalho, Javier F. Cortés-Martinicorena, James Deschner, Gail V. A. Douglas, Rodrigo A. Giacaman, Vita Machiulskiene, David J. Manton, Daniela P. Raggio, Francisco Ramos-Gomez, Ruxandra Sava-Rosianu, Natalia S. Morozova, Gianrico Spagnuolo, Ana Vukovic, Thomas G. Wolf

**Affiliations:** 1Department of Restorative, Preventive and Paediatric Dentistry, University of Bern, Freiburgstrasse 7, 3012 Bern, Switzerland; marcela.betancourt@zmk.unibe.ch (M.D.-B.); thiago.saads@zmk.unibe.ch (T.S.C.); thomas.wolf@zmk.unibe.ch (T.G.W.); 2Department of Surgery, Microsurgery and Medicine Sciences, School of Dentistry, University of Sassari, Viale San Pietro, 07100 Sassari, Italy; 3Department of Biomedical, Surgical and Dental Science, University of Milan, Via Beldiletto 1, 20142 Milan, Italy; maria.cagetti@unimi.it; 4Faculty of Medicine and Dentistry, UCLouvain, 1200 Brussels, Belgium; joana.carvalho@uclouvain.be; 5Private Practice, 31001 Pamplona, Spain; javiercortes@dentalcortes.es; 6Department of Periodontology and Operative Dentistry, University Medical Center of the Johannes Gutenberg University Mainz, 55116 Mainz, Germany; James.deschner@uni-mainz.de; 7Department of Dental Public Health, School of Dentistry, University of Leeds, Leeds LS2 9JT, UK; g.v.a.douglas@leeds.ac.uk; 8Cariology and Gerodontology Units, Department of Oral Rehabilitation, Faculty of Health Sciences, University of Talca, 3460000 Talca, Chile; giacaman@utalca.cl; 9Clinic of Dental and Oral Pathology, Faculty of Odontology, Lithuanian University of Health Sciences, 44131 Kaunas, Lithuania; Vita.Maciulskiene@lsmuni.lt; 10Paediatric Dentistry, Centrum voor Tandheelkunde en Mondzorgkunde, UMCG, University of Groningen, 9700-9747 Groningen, The Netherlands; d.j.manton@umcg.nl; 11Department of Orthodontics and Pediatric Dentistry, School of Dentistry, University of Sao Paulo, 05508-060 Sao Paulo, Brazil; danielar@usp.br; 12UCLA Center for Children’s Oral Health (UCCOH) UCLA School of Dentistry, Los Angeles, CA 90095-1668, USA; frg@dentistry.ucla.edu; 13Department of Preventive, Community Dentistry and Oral Health, Faculty of Dentistry, University of Medicine and Pharmacy “Victor Babes” Timisoara, 300363 Timisoara, Romania; savarosianu@yahoo.com; 14Institute of Dentistry, Sechenov University, 3200900 Moscow, Russia; kns74@bk.ru (N.S.M.); gianrico.spagnuolo@gmail.com (G.S.); 15Department of Neurosciences, Reproductive and Odontostomatological Sciences, University of Naples “Federico II”, Via Pansini 5, 80131 Naples, Italy; 16Department of Pediatric and Preventive Dentistry, School of Dental Medicine, University of Belgrade, 11000 Belgrade, Serbia; ana.vukovic@stomf.bg.ac.rs

**Keywords:** COVID-19, infection, dentist, protective measures, awareness, infection control

## Abstract

The Centres for Disease Control and Prevention and the World Health Organization have developed preparedness and prevention checklists for healthcare professionals regarding the containment of COVID-19. The aim of the present protocol is to evaluate the impact of the COVID-19 outbreak among dentists in different countries where various prevalence of the epidemic has been reported. Several research groups around the world were contacted by the central management team. The online anonymous survey will be conducted on a convenience sample of dentists working both in national health systems and in private or public clinics. In each country/area, a high (~5–20%) proportion of dentists working there will be invited to participate. The questionnaire, developed and standardized previously in Italy, has four domains: (1) personal data; (2) symptoms/signs relative to COVID-19; (3) working conditions and PPE (personal protective equipment) adopted after the infection’s outbreak; (4) knowledge and self-perceived risk of infection. The methodology of this international survey will include translation, pilot testing, and semantic adjustment of the questionnaire. The data will be entered on an Excel spreadsheet and quality checked. Completely anonymous data analyses will be performed by the central management team. This survey will give an insight into the dental profession during COVID-19 pandemic globally.

## 1. Introduction

The novel coronavirus SARS-CoV-2 (severe acute respiratory syndrome coronavirus 2) pandemic has affected the world deeply. COVID-19, as the disease has become known, is the third coronavirus to emerge in the human population recently, preceded by the SARS-CoV (severe acute respiratory syndrome coronavirus) outbreak in 2002 and the MERS-CoV (Middle East respiratory syndrome coronavirus) outbreak in 2012. Organizations such as the Centers for Disease Control and Prevention (CDC) and the World Health Organization (WHO) have developed preparedness and prevention checklists regarding the containment of the spread of COVID-19, to be used by public and general healthcare professionals [1,2]. The SARS-CoV-2 human-to-human transmission is via respiratory and saliva droplets or direct contact with cases or with contaminated surfaces [3]. Airborne transmission of the virus might occur during medical procedures that generate aerosols, even if this transmission route is not yet fully clarified [2]. Avoiding close contact (less than 1 m) with people, especially those with positive tests and or respiratory symptoms, is one of the most important preventive measure to be taken to prevent the spread of the infection. Having in mind the worldwide spread of SARS-CoV-2 from China to all other parts of the world [2,4], it is of utmost importance to design feasible preventive strategies in dental settings. The initial outbreak in Wuhan spread rapidly, affecting other parts of China. Cases were soon detected in several other countries. As of 22 July 2020, 14,562,550 cases of COVID-19 (in accordance with the applied case definitions and testing strategies in the affected countries) have been reported, including 607,781 deaths [2]. Different numbers of cases have been reported around the world, in South-East Asia (1,478,141 cases), Europe (3,103,674 cases), Eastern Mediterranean (1,400,544 cases), Western Pacific (266,190 cases), Africa (611,185 cases), and the Americas (7,702,075 cases) [2,4].

Dental treatments for patients with COVID-19 or suspected to be infected by the virus are suggested to be postponed, except in case of urgent treatments; nevertheless, undiagnosed infected subjects without or with very mild symptoms might be seen for dental treatment. Furthermore, having in mind that many dental offices around the world have returned to providing routine or not urgent dental care, limited knowledge and awareness, unavailability of protocols and tests, and ineffective personal protective equipment (PPE) use might lower the level of safety of patients, dentists, and dental care workforce, increasing the infection spread in the community [5].

The risk of cross-infection in dentistry is considered to be high [6,7], since splatters and aerosols produced during routine dental treatments, combined with the physical proximity to the patient’s face, increase the risk [8]. This issue might be an important and unacceptable professional hazard when infective agents, such as coronaviruses, are widespread in the population [6]. Dentists and health care professionals working in wards with pneumonia patients are at higher risk of developing infective diseases during their regular activities [7]. Data on the real risk of virus dispersion by dental procedures are urgently needed, since none are available in the literature currently [7,8]. In a recent paper, the stability of SARS-CoV-2 and SARS-CoV-1 in aerosols and on various surfaces was investigated in experimental conditions, showing that airborne transmission of SARS-CoV-2 is plausible, since the virus can remain viable and infectious in aerosols for hours [8,9,10]. Without data on airborne SARS-Cov-2 transmission from actual dental care situations, operational envelopes and disinfection procedures to prevent cross-infection are plausible, but hypothetical. Thus, extreme precautions appear to be necessary.

Well-designed questionnaires are a useful method to collect data easily from participants in studies [11]. Questionnaires to investigate dentists’ knowledge, attitudes, and perceptions regarding viral infection control in the dental environment show that awareness and precautionary measures carried out by dentists on patients with a viral infection are not always completely satisfactory [10,11,12,13,14,15]. Both the risk perception by dentists regarding the SARS-CoV-2 infection and the protective measures they took during the lockdown and at work restart, in countries where non-urgent dental treatment has been suspended, are speculative and scarcely investigated.

The aim of the present protocol will be to evaluate the impact of the COVID-19 outbreak among dentists working in different countries with various levels of prevalence of the pandemic.

Research questions.
What is the prevalence of the symptoms/signs reported by dentists worldwide presumably referable to the COVID-19?What is the level of preparedness regarding protective measures and PPE to reduce the risk of viral transmission?What is the level of awareness and risk perception of dentists regarding COVID-19?

## 2. Methods

### 2.1. Study Design

The survey is designed as a cross-sectional survey using a previously standardized questionnaire.

### 2.2. Time Period

March 2020-November 2020 (Gantt Chart Figure 1).

### 2.3. Study Settings

The central management team contacted 35 collaborating research groups around the world. The countries participating in the survey are shown in Figure 2.

### 2.4. Study Population

Registered dentists working in national health systems, working in private or public clinics, including general or specialist dentists will be enrolled. Participants who cannot communicate in the vernacular of the translated questionnaire will be excluded.

### 2.5. Sample Size Calculation

The survey will be conducted on a convenience sample of countries/areas. In each country/area, the total number of working dentists will be ascertained and a high (5–20%) proportion of dentists will be invited to participate. Countries/areas in which the requested sample size are not reached will be included in the main results of the survey.

### 2.6. Development and Refinement of the Questionnaire

The first group of items included in the questionnaire will be related to the health situation, risk, and knowledge of an infectious disease, derived from a questionnaire developed for the SARS risk [8], following the Stehr–Green scale to build up the questionnaire [10]. The questionnaire has four domains: the first regarding personal data (age, gender, area of living and working, working status); the second regarding health conditions (symptoms/signs related to COVID-19); the third on working conditions and PPE adopted after the outbreak of the infection; and the fourth regarding the knowledge and the self-perceived risk of infection. Among the PPEs included in the questionnaire, some, such as the use of sterile gloves, do not have a scientific justification, but they were deliberately inserted to check whether the answers were selected with the sole logic of demonstrating that any contrast measures regarding the virus had been implemented or whether the equipment adopted is the result of thoughtful decisions.

The evaluation methodology of the questionnaire will include two stages for each country:Stage I (Questionnaire translation and testing):

Translation of the questionnaire from the original English version into the different languages will be performed by a researcher from each research group. The researcher, with expertise in public health dentistry, must have very good English skills, and he/she will have to determine its conceptual equivalence in his/her language.

Back-translation from the different language into English by a translator who does not belong to the research team will be performed.

Stage II (Pilot test and semantic adjustment of questionnaire)

A sample of dentists working in the respective country will be selected.

The sample will be divided randomly into two sub-samples: one for the pilot study to check semantic comprehension, and another larger group of dentists will be recruited for the subsequent validation study.

The translated questionnaire will be administered to the pilot sub-sample.

To determine its reliability, the questionnaire will be re-sent to the pilot sample a second time 4–7 days after the first administration, without any recommendations from the researchers.

The questionnaire will be adjusted in accordance with findings from the tests described above.

An online survey (Survey Monkey™ SVMK Inc., San Mateo, CA, USA or REDCap Research Electronic Data Capture https://www.mc.vanderbilt.edu, or Google Form google.com/forms/about/or similar online platform) will be prepared.

### 2.7. Coordination and Survey Participation Sites

The research team in each country/area will be required to prepare the survey (see above) and oversee data collection and management locally. Local collaborators may add questions related to their country/area or for specific research reasons and will publish them after the main collaborative publication. Each research team will be free to reach dentists through the channel(s) they deem most appropriate and effective in their reality to involve the highest number of participants, such as professional orders, scientific societies, or Facebook groups.

The research team of each country/area will be specifically responsible for:Obtaining local audits, or research ethics approval (IRB/HREC approval).Identifying dentists who will be invited to participate in the survey.Deciding on which platform the survey will run in their country, how to reach the dentists, and when to run the survey. The platform for the survey will be set to avoid duplicated answers. The duration of the survey in each country will be for, at least, seven days.Accuracy and any misconduct related to their research project.Supporting translation of the questionnaire (both forward and reverse translations) into the local language and conducting the pilot test and semantic adjustment of the questionnaire.Following all steps of the local survey.Verifying that the data are accurately collected and organized, before sending them to the central management team (University of Bern).Writing a report about the country-based local data.

### 2.8. Data Management

The data will be entered into an Excel (Microsoft Corp., WA, USA) spreadsheet and quality-checked by a researcher to ensure accuracy. Each survey response will be completely anonymous; the questionnaire will have to explicitly avoid any identification of participants’ identities. Only the management team will be able to access all data. Data from all involved countries/areas will be exported into Excel™ 2019 for Mac (or Windows equivalent); the data will be then cleaned and transferred in STATA16™ (Statacorp, TX, USA) for their statistical analysis. Data analyses will be centrally performed by the central management team.

### 2.9. Ethical Approval

Each research team will apply for ethical approval according to the regulations and law of the relevant country. It is possible that some countries do not need ethical approval or that using approval from elsewhere meets the regulations. All the respondents of the survey will complete an informed consent question embedded on the first page of the questionnaire. If the participant answers “YES” to the first question of the form, he/she automatically agrees to participate and will begin the survey. By using the skip-logic survey method, users who disagree with the informed consent question will be directly conducted to the end of the survey. No participant will be forced to participate in the survey, and their participation will be based on their agreement that could be withdrawn at any time. All participants have the right to leave one or more specific questions unanswered or withdraw from the survey. In addition, individual answers to the questionnaire will be inaccessible.

### 2.10. Confidentiality and Data Retention

The responses collected through this survey will be strictly anonymous and confidential; no identification of either participants or health centres will be possible. Individual responses are not of interest; the collective and combined outcomes derived from each participant country will be reported at an aggregate level.

The collected, electronic data will be stored on a password protected and backed-up computer drive and remain confidential—only authorized team members will have access to it. In addition, data will be completely encrypted and coded for use mainly in statistical analysis using computer software.

## 3. Discussion

Infection prevention and control during health care is always recommended, especially when COVID-19 infection is suspected (WHO 2020a). Up to now, there has been no consensus on the provision of dental services during the epidemic of COVID-19. A guideline was recently published by the CDC [16], and in each country there have only very recently been multiple sources of guidance from professional bodies; however, these conflict with one another and, therefore, are unhelpful. At the current stage of the pandemic and probably still for several months, dentists should utilize strict personal protection equipment and avoid or minimize operative procedures which can produce droplets or aerosols. The management of the operating area for dental care should be quite similar to those where patients affected by infectious and highly contagious diseases are treated, depending on the epidemiologic situation in each country/region. It is clear from all sources of guidance that, as often as possible, staff should work at an adequate distance from patients; furthermore, there are suggestions that handpieces should be equipped with non-return devices to avoid contamination, decreasing the risk of cross-infection. Finally, the dentist should favor procedures which reduce the quantity of aerosol dispersed into the environment [4,6]. The philosophy of minimal intervention dentistry could be adopted, helping to reduce the spread of viral particles during dental care. Conservative approaches, such as the atraumatic restorative technique, partial or selective removal of carious tissues with hand instruments or manual scaling of calculus, should therefore be preferred, when possible.

Individual prevention, both for health personnel and for patients, must be associated with the prevention of the spread of the virus through environmental remediation. In particular, due to the high proliferation of the virus in the droplets and aerosols exhaled by coughing and sneezing, every surface in the waiting room should be considered at risk; therefore, in addition to providing adequate periodic air exchange, all surfaces, chairs, and doors that come into contact with health care professionals and patients must be considered “potentially infected”. It may be useful to provide masks and disinfectants to patients for hand application or hand washing while they stay in waiting rooms.

The data collected in this study will play an important role in raising awareness among professionals and policymakers on the impact of the COVID-19 pandemic on the dental profession, but also it will assist preparedness for potential future viral outbreaks, including devising research agendas. Effective control during early stages of a pandemic will play a crucial role in preventing viral spread and outbreak in other cities/countries. Identifying and analyzing epidemiological issues among dentists, communicating international experiences, and disseminating transparent and usable results will provide scientific data, empower decision-making, and reduce community transmission during current pandemic and post-crisis times.

The results of this project will help to define the best strategy to organize the dental workforce. This study represents an interesting and rather unusual experience for the dental profession globally to join efforts to obtain important information from more than 30 countries, with different socio-politic contexts, geographical and climate conditions, as well as cultural particularities. The final goal is to be able to care for all patients while minimizing the risk to staff, thus maintaining a healthy caregiver workforce. The methodological ease of this strategy makes it appealing and reproducible. Similar initiatives can be approached in the near future.

## Figures and Tables

**Figure 1 ijerph-17-05598-f001:**
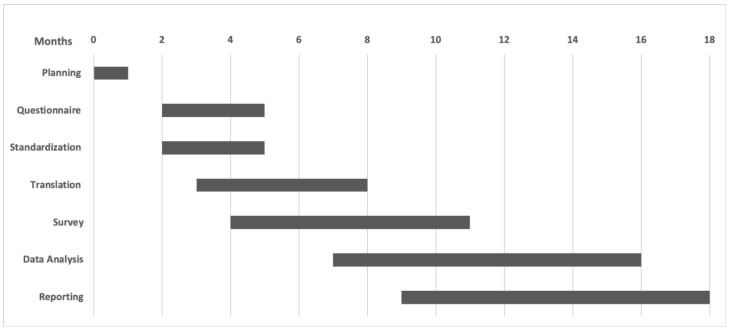
Gantt Chart of the survey.

**Figure 2 ijerph-17-05598-f002:**
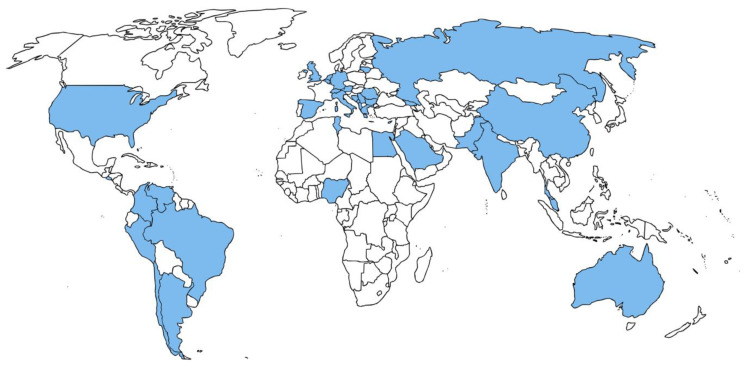
Participating countries.

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
