# Peer review of "Study Protocol for an Online Questionnaire Survey on Symptoms/Signs, Protective Measures, Level of Awareness and Perception Regarding COVID-19 Outbreak among Dentists. A Global Survey"

_ijerph, 2020, doi:10.3390/ijerph17155598_

Round 1

Reviewer 1 Report

-The manuscript is a proposal to develop a study protocol to evaluate the impact of the COVID-19 outbreak among dentists working in different countries with various prevalence of the pandemic. The proposal is collaborative effort among academics and clinicians from many countries and with a diversity of specializations.

-The framework is basically sound and the survey when completed is likely to be useful to all dentists.

Author Response

We really thanks the reviewer for the nice support to our study.

Reviewer 2 Report

The authors present a study protocol for a cross-sectional survey among dentists in some 20 countries worldwide. The aim is (i) to assess the prevalence of the symptoms/signs referable to the pandemic in dentists worldwide (ii)  the level of preparedness regarding preventive measures and PPE to reduce the risk of viral transmission? and (iii) to assess the risk perception and awareness level of dentists regarding the new coronavirus?

This is a relevant question of interest not only to dentists but to health professional in general. Therefore, an appropriate survey is recommended.

However, the study protocol given has a number of shortcomings and lacks necessary detail. It is planned to recruit a convenience sample. This method has the well-known drawbacks that the group of participants is highly selected. I would assume that participation is correlated with the general attitude towards the pandemic which would make any firm conclusions from the survey problematic.   

It is stated that “a high (5%-20%) proportion of dentists will be invited to participate“.  What does this mean? In the US there are some 200.000 dentists. Is it planned to contact 10.000-40.000?  In China and in India the numbers are even higher. How will they be contacted? by mail / email? is a full electronic list available in all participating countries? Which response rate assume the authors?

There is no information on envisaged the total number. What is the aim to achieve as a sample size?  

In my view it would be preferable to collect a moderate number, randomly selected, and perhaps offer an incentive to increase the response rate.

The time frame suggested is March 2020-November 2020 (presumably meant: 2021). This has to be updated, unless the study is already ongoing (in which case, of course, any suggestion to modify the design is useless).

Reviewer 3 Report

The aim of this work is to evaluate the impact of the COVID-19 outbreak among dentists working in different countries with various prevalence of the pandemic, on the prevalence of the symptoms/signs referable to the pandemic in dentists, the level of preparedness regarding preventive measures and PPE to reduce the risk of viral transmission and  the risk perception and awareness level of dentists regarding the new coronavirus.

This lies within the scope and objectives defined by the IJERPH. It is an original subject and presents scientific and clinical relevance with originality. The base references for the investigation are well identified but could be improved.

The title is appropriate and the key words too.

The text must be reviewed in order to answer some of the questions addressed.

The abstract is in line with the manuscript.

The introduction is well structured and within the scope of the work. Summarizes recent research/information related to the topic. Although it would be necessary to:

- clarify that the aerosol transmission is not yet proved (line 62);

- no need of describing all countries data, as it is a daily change, so it would be needed to highlight the Continents data;

- patients with COVID may be treated with adequate protection measures and environment (line 82);

- there are a significative number of protocols/guidelines throughout the world, so I wouldn’t say that “there are limited available protocols” (line 85);

- clarify what you mean with “low resource setting” (line 86);

- in a world context it would be better to have “ dentists and dental care workforce” (line 92);

The objective is clear, although the answer to question a) seems to be impossible to achieve with the present methodology.

The methodology is clear. The convenience sample is always more limited in terms of outcomes. Nevertheless, it should be clear that:

- you might end up without having representative samples in many participating countries;

- although is each local team responsibility, it should be clear that the platform for the survey should be able to avoid duplicated answers;

The discussion is interesting, but there are some additional notes;

- it is missing the reference to ECDC guidelines (line 219);

- it should be clear that the preventive measures should be adequate to the epidemiologic situation of the country/local (line 223);

- it is missing the previous mouthwash before starting treatment (line 225);

Further comments are in the PDF manuscript.

Based on these comments I think the article should be accepted with a minor review.

Round 2

Reviewer 2 Report

I am not too optimistic about the participation rate, however you may be able to prove the contrary. Therefore I think the conclusions that can be drawn from this study will probably be limited. However the COVID-19 pandemic requires immediate action, therefore it is appropriate to perform such a study.